# Effect of Yogurt Enrichment with Wood Tannin during Shelf Life: Focus on Physicochemical, Microbiological, and Sensory Characteristics

**DOI:** 10.3390/foods12020405

**Published:** 2023-01-14

**Authors:** Negin Seif Zadeh, Martina Tedesco, Sofia Basso, Daniela Ghirardello, Samuele Giovando, Michele Battaglia, Giuseppe Zeppa

**Affiliations:** Department of Agriculture, Forest and Food Sciences (DISAFA), University of Turin, 10095 Grugliasco, Italy

**Keywords:** yogurt, tannin, polyphenols, antioxidant activity, functional food

## Abstract

Six food-grade tannins obtained from different woods were used as a source of polyphenolic compounds at two concentrations (0.5% and 1% *w*/*w*) in yogurt formulations and monitored during 3 weeks of storage. Yogurt containing tannins showed significantly higher total phenolic content (+200%), antioxidant activity (+400%), and syneresis (+100%) than control. These changes were higher with fortification at 1%. Tannin origin also significantly influenced the yogurt composition and yogurt obtained from a Turkish gall showed higher values of total phenolic content (4 mg GAE/g) and antioxidant activity (17 μM Trolox/g). Yogurt color was evaluated by CIELab parameters, and their values were influenced by tannin origin and concentration. The addition of tannins did not significantly affect the number of lactic acid bacteria. Yogurt with a lower amount of tannins (0.5% *w*/*w*) received higher consumer acceptability but significant differences in preferences were due to tannin origin. In particular, yogurt added with tannin obtained from Quebracho wood at 1% *w*/*w* showed higher consumer preference. The obtained results would provide an opportunity for dairy producers to develop a novel dairy food with high nutritional quality.

## 1. Introduction

Tannins are secondary metabolites present in various parts of the plant, such as the buds, bark, wood, roots, fruits, and seeds [1,2]. They are produced under stress as defensive agents to protect plants against environmental hazards, and they have been found to be linked with biological activities such as antioxidant, antimicrobial, anti-inflammatory, antidiabetic, and cardioprotective activities [3,4,5]. Chemically, tannins are a phenolic group of compounds characterized by heterogeneous structural properties reflected by a wide range of molecular weights, varying from 500 to 20,000 Da [6]. Similar to other phenolic compounds, tannins exert free radical scavenging activity and protect against lipid peroxidation due to their structure since they possess phenolic rings that are able to donate electrons and trap unstable radicals [7,8]. According to their structure, tannins are classified into four main groups: hydrolysable tannins (present in small amounts in plants and, based on the products obtained from their hydrolysis, are divided into gallotannins and ellagitannins), condensed tannins (also known as proanthocyanins, which include more than 90% of the commercial production of tannins), complex tannins (which have higher molecular weights and are produced from the linkages between condense tannins with ellagitannins and/or gallotannins) and 4-phlorotannins (belonging to the brown algal species) [4,9]. In plants, the portions that contain significant amounts of tannin are either very small and edible or not edible at all (for example, thin brown skin or an outer fibrous husk similar to that of the hazelnut) [10]. For this reason, tannins are mostly applied in nonfood industries, such as leather tanning and wood adhesive production. Recently, due to the evidence of the physiological value of these compounds and the increasing consumer demands for functional food on the market, studies on new relevant applications in the food sector are rising [4]. Tannins have been proposed as food preservers because they improve the safety and shelf life of products. Tannins can contribute an astringent or bitter taste, which gives some characteristics to beverages, such as red wines and beers [11]. The sensation apparently results from the interaction between tannin constituents and proteins in the saliva and/or the mucous tissue of the mouth. They have also been used as clarification agents in beverages because they can bind and precipitate a range of molecules. However, the use of tannins as an ingredient or additive in the food industry is still limited. One of the reasons is that they can sometimes be linked to unpleasant organoleptic properties in the final product. Yogurt is already considered to be a healthy food because it contains viable probiotic bacteria; however, it does not contain phenolic antioxidant compounds. Thus, the objective of this study was to evaluate the incorporation of plant-based tannins, extracted from wood, in yogurt to obtain a novel dairy food rich in polyphenols with functional activities. Wood tannins are inexpensive, stable during processing, compatible with food, and their addition to the food matrix can be effective even at low concentrations. To this end, six commercial food-grade tannin wood extracts (TWE) were added to the stirred yogurt, and the physicochemical and sensory properties were monitored during 21 days of refrigerated storage.

## 2. Materials and Methods

### 2.1. Materials

All chemicals used were of analytical grade. Folin–Ciocalteu phenol reagent, 2,2-diphenyl-1-picrylhydrazyl (DPPH), 6-hydroxy-2,5,7,8-tetramethylchroman-2-carboxylic acid (97%; Trolox), methanol (99.9%), sulfuric acid (98%), acetonitrile, formic acid (98–100%), ethanol (99.9%), Carrez I, and Carrez II were provided by Sigma-Aldrich (Milan, Italy). Ultrapure water was prepared using a Milli-Q filter system (Merck Life Science SrL, Milan, Italy).

Six commercial food-grade tannin wood extracts were furnished by Silvateam S.p.A (Cuneo, Italy): WellTan NN, WellTan QS-SOL, WellTan FNG/R, WellTan C, WellTan ARB, and WellTan QF. WellTan NN and WellTan QS-SOL are two types of Quebracho wood tannins from *Schinopsis lorentzii* and *Schinopsis balansae*, WellTan FNG/R is a Turkish gall tannin from *Quercus infectoria*, WellTan C is a Tara pod tannin from *Caesalpinia spinosa*, WellTan ARB is a chestnut wood tannin, and WellTan QF is a mixture of quebracho wood tannin and Turkish gall tannins. The tannins were furnished as powder. The stirred yogurt (dry extract 12.44%, fat 3.38%; protein 3.78%; carbohydrates 4.59%) was provided by “Terra Nuova” (Cuneo, Italy).

The six tannins were added to the stirred yogurt at two concentrations: 0.5% and 1% *w*/*w*. The yogurts were mixed completely and kept at 4 °C for 21 days. After production, the analyses were carried out on Days 1, 7, 14, and 21. A plain stirred yogurt was used as a control.

### 2.2. Physicochemical Analysis

A Crison Microph 2002 pH meter (Crison Strumenti SpA, Carpi, Italy) was used to evaluate the pH of the samples. Titratable acidity was determined according to AOAC [12]. Briefly, 10 g of the sample was diluted with ultrapure water up to 100 mL. Titration was carried out using 0.1 N NaOH until a pH of 9.2 was reached. The titratable acidity was expressed as % of lactic acid calculated with the following formula:$$\mathrm{Lactic}\;\mathrm{acid}\;(\%)=\;\frac{\left(\mathrm{mL}\;\mathrm{NaOH}\;0.1\;N\;\times 0.9\right)}{\mathrm{sample}\;\mathrm{weight}}$$

The syneresis of yogurt was measured as described by Bertolino et al. [13]. Briefly, 20 g of yogurt was centrifuged at 350× *g* for 30 min at 10 °C, and then the whey was removed. Syneresis was expressed as the volume of whey per 100 mL of yogurt.

Color was evaluated through the CIELAB color space using a CM-5 spectrocolorimeter (Konica Minolta, Tokyo, Japan) in transmittance and Specular Component Excluded (SCE) modality. The color space parameters L*, a*, and b* (CIELAB values) were used to measure the colorimetric characteristics, where L* is a coefficient of lightness ranging from 0 (black) to 100 (white), a* indicates the red-green colors (red with +a*; green with −a), and b* represents the yellow-blue colors (yellow when positive b*; blue when negative b*). All analyses were performed in triplicate.

### 2.3. Polyphenol Extraction

For extracting total phenolic compounds (TPCs) from yogurt, the technique proposed by Vásquez et al. [14] was adopted with some adjustments. Briefly, 8 g of yogurt was added to 10 mL of 50% methanol and mixed for 20 min on a VDRL 711 orbital shaker (Asal s.r.l., Milan, Italy) at a constant oscillation (1.67 oscillations/s) at ambient temperature (20–22 °C). Then, 500 µL of Carrez I (15% potassium ferrocyanide trihydrate), 500 µL of Carrez II (30% zinc acetate heptahydrate), 5 mL of acetonitrile, and 1 mL of 50% methanol were added and mixed well by vortexing for 1 min after each step. To achieve complete clot protein precipitation, the mixture was then allowed to stand for 25 min and subsequently centrifuged at 7800× *g* for 15 min at 5 °C. The supernatant was filtered through a 0.45 μm pore size cellulosic membrane syringe filter. The extractions were performed in triplicate for each sample, and the extracts were stored at −18 °C in amber glass vials until further analyses.

### 2.4. Determination of Total Phenolic Content

The total phenolic content (TPC) of the extract was estimated according to the procedure described by Singleton, Orthofer, and Lamuela-Raventos [15] by using 96-well microplates and a BioTek Synergy HT spectrophotometric multidetection microplate reader (BioTek Instruments, Milan, Italy). Twenty microliters of opportunely diluted sample extract were dispensed into the corresponding well of a microtiter plate with 100 µL of Folin–Ciocalteu aqueous reagent (diluted tenfold). After the content was mixed and left for 3 min, 75 µL of a saturated sodium carbonate solution was added to each well. The solution was mixed again, and the plate was incubated for 1 h at 25 °C in the dark. The absorbance was recorded at 740 nm against a reagent blank. For quantitative purposes, a calibration curve was prepared for quantification using gallic acid as a stable phenolic compound (100–500 µM; R^2^ = 0.9994). The amount of the total phenolic compounds was expressed as mg of gallic acid equivalents (GAE) per g of sample. All determinations were performed in triplicate.

### 2.5. Determination of Radical Scavenging Activity

The DPPH radical-scavenging activity (RSA) was performed on extract according to von Gadov et al. [16] with some modifications. For the modified procedure, a 120 µM solution of DPPH^•^ was prepared in 80% ethanol daily. Twenty microliters of the previously diluted extract and 180 μL of a DPPH^•^ solution were added to each of the 96 microplate wells. The plate was then vigorously shaken, covered, and left to stand in the dark for 30 min at 25 °C. The decrease in DPPH^•^ absorbance was measured at 517 nm against a reagent blank, and the inhibition percentage (IP) of the radical was calculated according to the following equation:$$\mathrm{IP}\;(\%)=\left[\;\frac{\left(A0-A30\right)}{A0}\;\right]\times 100$$
where A0 is the absorbance at the initial time and A30 is the absorbance at 30 min. Trolox was used as a standard at 12.5–300 µM to construct a calibration curve (R^2^ = 0.9982). The radical scavenging activity values of each sample were expressed as μM Trolox equivalents (TE) per g of sample. All samples were assessed in triplicate and averaged.

### 2.6. Sugar and Acid Determination

Organic acids and sugars were evaluated by HPLC-DAD-RI analysis. Briefly, 20 mL of 0.013 N H_2_SO_4_ (mobile phase) were added to 5 g of yogurt samples and mixed for 20 min with a horizontal shaker (PBI, Milano, Italy) at 120 rpm. The slurry was subsequently centrifuged for 10 min at 6000× *g* and 10 °C, and the supernatant was filtered through 0.45 and 0.20 μm pore size cellulosic membrane syringe filters and used for HPLC analysis. The HPLC system (Thermo Electron Corporation, Waltham, MA, USA) was equipped with an SCM 1000 degasser, a P2000 binary gradient pump, a multiple autoinjector AS3000, a photodiode array UV6000LP, and a refractive index detector RI-150. The detectors were connected in series. Acidified water (0.013 N H_2_SO_4_) was used as the mobile phase, and the isocratic elution method was applied with a flow rate of 0.6 mL/min. The analysis was performed using a reverse phase Aminex HPX-87H column (300 mm × 7.8 mm) equipped with a Microguard cartridge (Bio-Rad Laboratories, Hercules, CA, USA) working at 65 °C. The data were elaborated by a ChromQuest 5.0 chromatography data system (Thermo Electron Corporation, Waltham, MA, USA), and quantification was performed using calibration curves from the injection of analytical grade external standards obtained under the same conditions.

### 2.7. Bacterial Count

Enumeration of lactic acid bacteria (LAB) was carried out using traditional microbiological methods (colony-forming unit) [17] and the yogurt samples were diluted with Ringer’s solution (Merck). *Streptococcus* spp. were counted on M17 agar (Merck, pH 7.1), while MRS agar (Merck, pH 5.4) was used for enumeration of *Lactobacillus* spp. The plates were incubated at 37 °C for 48 h under anaerobic conditions and reported as the log of cfu/g.

### 2.8. Liking Test

The liking test was conducted with 50 adult subjects (females = 70%, age range: 24–64 years) who were recruited from the staff of the University of Turin. Written informed consent was obtained from all participants before the test. Participants received individual plastic cups and rinsed their mouths with noncarbonated water before beginning the evaluation. Participants tasted the samples according to the tray presentation order and were blinded without any information about the innovativeness of the yogurt to avoid a potential effect on the liking scores. A nine-point hedonic scale test was used to evaluate appearance, odor, taste, flavour, texture, and overall liking of the yogurt, where 1 represented “extremely dislike” and 9 represented “extremely like” [18]. The tests were performed in an air-conditioned room with white light at approximately 21 °C. Sensory assessment of yogurt samples was performed after 21 days of cold storage. Samples were served at 10 ± 1 °C.

### 2.9. Statistical Analysis

All data analyses were performed with SPSS 28.0 Package (SPSS Inc., Chicago, IL, USA). Analysis of variance (ANOVA) was used to determine significant differences among the results (means ± standard deviations); the significant difference and mean values were obtained by Duncan’s test at a confidence level of *p* < 0.05.

## 3. Results and Discussion

### 3.1. Physicochemical Characterization

The data obtained from the pH, titratable acidity, and syneresis measurements of the yogurts during cold storage are reported in Table 1. Significant differences were observed for the pH with respect to storage time and tannin type. The pH dropped slightly during the storage for all the tannin-added samples as well as the control yogurt. While, in the first day of production, tannin addition did not affect the pH among samples significantly, it led to increased acid production in yogurts at the end of storage. The highest acidification was found in the yogurt with 1% tannin C, where the pH decreased from 4.15 ± 0.01 in the control to 3.87 ± 0.07 in the experimental yogurt. Although the highest amount of fermentation occurs in the first hours after the addition of the starter, the fermentation bacteria continue their activity partially during the storage period. The lower pH of the tannin-added yogurt at the end of the storage can be related to the difference in the fermentation medium [19]. The reduction in pH during storage coincided with the increase in titratable acidity, suggesting the continued production of organic acids during storage. While the acidity of the control sample changed from 0.61 ± 0.01 g lactic acid/g yogurt at the day of production to 0.87 ± 0.02 g lactic acid/g yogurt at the end of storage at Day 21, the maximum increase in acidity was found in yogurt supplemented with 1% tannin C, changing from 0.83 ± 0.01 to 1.19 ± 0.02 g lactic acid/g yogurt. An increase in acidification after yogurt supplementation was reported in various studies [18,19]. The syneresis of all yogurts with added tannins was higher than that of the control sample. Syneresis decreased slightly during cold storage, and the type and amount of tannin significantly affected the separation of the whey. The highest syneresis was observed for sample QS-SOL with 1% tannin, both on the production day (44.87%) and at the end of the shelf life (38.71%). The yogurt syneresis was considerably influenced by the addition of tannin; as in the enriched yogurts, the amount of expelled serum was approximately twice that of the control sample. An increase in the syneresis of yogurt after the addition of fruits and flavors has been reported in other studies, and a similar trend was observed when yogurt was supplemented with grape pomace, sour cherry, and Mexican caramel jam [18,20,21]. Gilbert et al. [22] explained that shearing and the mechanical forces on the milk coagulum in post-fermentation unit operations can result in a reduced viscosity and increased serum separation. Considering that, in this study, tannins were added at the end of the fermentation stage when the yogurt texture had already formed, the addition of tannins and subsequent stirring until a homogenous texture appeared led to a decrease in coagulation and an increase in the separation of the whey.

### 3.2. Color Evaluation

The color of yogurt is a key factor in consumer acceptance and marketing. In the production of functional yogurt, attention should also be paid to its attractiveness, in addition to its nutritional properties, for the customer. The color of polyphenolic compounds such as tannins is susceptible to change over time. For this reason, the color properties of yogurt were monitored during the storage period (Table 2). By increasing the tannin concentration in yogurt, the lightness parameter L* was reduced, hence for all types of tannin, the sample with 0.5% of addition was lighter. L* was the highest for the control sample compared to the tannin-added yogurts, followed by FNG/R and QF, and the yogurt with 1% tannin C had the lowest L* on the first and last days of storage. L* values were increased at the end of storage for all samples except the control. This could happen due to the possible degradation of the phenolic compounds, and therefore, the lightness of the sample is increased. The same trend was reported for yogurts with edible flowers [23], turmeric, and blue pea natural phenolics [24]. Pires et al. [23] explained that lactic acid bacteria could negatively influence the stability of the colorant anthocyanins and phenols and intensify their degradation by producing destructive enzymes such as glycosidase. In the control sample with no tannin, the a* factor (red/green) was negative, and the b* factor (yellow/blue) was much lower in comparison with the tannin-added samples. As the tannin percentage increased, the redness and yellowness also increased. Unlike the lightness factor (L*), no visible color change was detected in tannin-added yogurts during storage.

### 3.3. Total Phenolic Content and Radical Scavenging Activity of Yogurt

The total phenolic content (TPC) values are shown in Table 3. The control yogurt sample had the lowest TPC (0.033 ± 0.001 mg GAE/g), which resulted from the natural polyphenols present in milk. These compounds are generally derived from eating and reducing compounds [25]. The addition of tannins significantly changed the phenolic content of the yogurt. On the first and last days of storage, the highest TPC was measured in the sample with 1% FNG/R tannin containing 4.49 ± 0.13 and 3.69 ± 0.03 mg GAE per g of yogurt, respectively. The sample with 1% addition of QS-SOL and QF was shown to have the second and third highest amount of phenolic compounds. Apart from tannin type, the amount that was added significantly affected the TPC of the yogurt, as samples with 1% tannin exerted twice the tannin content of the sample with 0.5% tannin. Among the fortified yogurts at the end of the shelf life, 0.5% C showed the lowest phenolic content (0.495 ± 0.025 mg GAE/g), which was still considerably higher than that of the control yogurt (0.039 ± 0.001 mg GAE/g). A higher TPC value was also reported in yogurt supplemented with grape [18], soursop, sweetsop, and custard apple [25] or mulberry [26], compared to the control stirred yogurt. There were significant differences (*p* < 0.05) in the DPPH scavenging activity among the tannin-added yogurt samples. The yogurt samples exerted scavenging activities in the descending order of FNG/R > QS-SOL > QF > ARB > NN and C > Control at 21 days of storage. As expected, the radical scavenging activity of the yogurts increased with increasing tannin concentration, while it decreased during cold storage, probably due to the degradation or oxidation of the phenolic compounds over time. At the end of the shelf life, the samples with 0.5% NN and C with 2.957 ± 0.117 and 3.059 ± 0.098 µM Trolox/g demonstrated the lowest scavenging ability among the tannin-added yogurts, while the control yogurt showed considerably lower activity (0.031 µM Trolox/g). In another study, Senadeera et al. [25] observed higher TPC and RSA in stirred yogurt after the addition of Annona species pulp as a natural source of tannins and fibers. Fidelis et al. [27] reported a concentration-dependent increase in the chemical antioxidant and reducing capacities of stirred yogurt by the addition of different concentrations of camu-camu (*Myrciaria dubia*) seed extract.

### 3.4. Organic Acids and Sugars

The changes in organic acids and sugars during 21 days of cold storage were monitored using the HPLC-DAD-IR technique. In total, seven organic acids (orotic acid, citric acid, pyruvic acid, lactic acid, uric acid, acetic acid, and butyric acid) and three sugars (lactose, glucose, and galactose) were detected in the yogurt samples. The addition of tannin and storage time did not significantly affect orotic acid (0.084 ± 0.002 g/100 g), uric acid (0.048 ± 0.001 g/100 g), and acetic acid concentrations (0.087 ± 0.003 g/100 g) (*p* < 0.05), while pyruvic acid decreased during the storage period from 0.050 ± 0.003 to 0.025 ± 0.001 g/100 g with no significant differences between tannin-added yogurts and the control. Pyruvic acid is produced due to carbohydrate and protein metabolism, and during fermentation, it is converted into lactic acid and other metabolites [28]. Glucose was found at very low levels of approximately 0.196 ± 0.011 g/100 g in the control sample on the first day of production, while it was not detected from Day 7 to the end of shelf life due to its consumption by LAB. Galactose was found at a level of 0.679 ± 0.011 g/100 g in the control yogurt and increased slightly up to 0.912 ± 0.009 g/100 g at the end of storage. During the fermentation of the starting yogurt culture, lactose is broken down into glucose and galactose. The fact that galactose is not metabolized by the micro-organisms of the yogurt starter usually results in the accumulation of this monosaccharide in yogurt during fermentation [29]. Although the levels of orotic, pyruvic, uric, acetic, butyric acids, glucose, and galactose were not influenced by tannin addition, and they showed the same trend as the control yogurt (data not shown), lactic and citric acids and lactose were affected by the type and concentration of the added tannins, as well as storage time (Table 4). Citric acid is the predominant organic acid in milk and is also present in yogurt. During cold storage, citric acid decreased in all samples, which confirms its role in bacterial metabolism. Costa et al. [28] explained that citric acid is the main substrate consumed by LAB in the production of acetoin and diacetyl. Lactic acid increased significantly for all samples during storage, where yogurt with 1% tannin C was the highest at the end of shelf life (18.06 ± 0.25 g/100 g). The lactic acid data correspond to pH and titratable acidity evolution. Lactose showed the opposite trend of lactic acid as it decreased in all yogurts during storage. This represents the typical behavior of LAB during fermentation in which the consumption of lactose in the starter culture results in the production of lactic acid as the main metabolite of fermentation. The lowest amount of lactose was found in the yogurt with 1% C (18.11 ± 0.49 g/100 g), which was found to be the most acidified sample. The obtained data from the measurement of acids and carbohydrates are in agreement with similar studies [28,30].

### 3.5. Viability of Lactic Acid Bacteria (LAB) during Cold Storage

Data in Table 5 represent the counts of *Streptococcus thermophilus* and *Lactobacillus delbrueckii* spp. *bulgaricus* (10^6^ CFU/g) in all yogurts after 21 days of cold storage. According to the Codex Alimentarius [31], the finished product must contain live lactic acid bacteria (LAB) in amounts higher than 10^7^ CFU/g at the end of the stated shelf life. In all samples, the total LAB charge is no less than 10 million per gram of product, which is the minimum requirement according to the Codex Alimentarius. While the type of added tannin influenced the viability of LAB at the end of the shelf life in comparison with the control sample, the concentration of the tannin addition did not affect their survival significantly. The total LAB counts of the QS-OSL and FNG/R yogurts were higher than that of the control, which suggests that they are suitable media for yogurt culture. Zahid et al. [32], after the addition of fruit peel powder to yogurt, reported an increase in the total LAB count after 28 days of storage. They explained that the proper amounts of phenolic compounds, along with dietary fibers, can contribute to the protection of yogurt culture from harsh conditions. Furthermore, some nondigestible condensed tannins may act as prebiotics and carriers for bacteria to prolong the viability of LAB. In another study, Kim et al. [33] found an increase in the number of LAB with increasing concentrations of lotus leaves added to stirred yogurt due to the abundant dietary fiber and tannins. Yogurt samples added with ARB, C, NN, and QF tannins had less *Streptococcus thermophilus* and *Lactobacillus delbrueckii* spp. *Bulgaricus* cells at the end of the shelf life compared with control, but as mentioned above, the final products contained enough living LAB cells to be claimed as yogurt.

### 3.6. Liking Test

The results of the liking test of the yogurts in terms of a 9-point hedonic scale are summarized in Table 6. In general, the high concentration of tannin negatively influenced the sensory attributes, and samples with higher amounts of tannin were less liked. Among tannin-added samples, yogurts with 0.5% FNG/R and 0.5% ARB received the highest scores, while the addition of 1% ARB resulted in the lowest scores for all sensory attributes. The appearance was the most preferred characteristic for all samples, and the yogurt containing FNG/R and QS_SOL received higher scores than the control sample. In the case of the aroma and texture, consumers gave scores above 5 to all the samples; thus, these attributes were strongly affected by the addition of tannins probably due to the astringency and bitterness of tannins. In agreement, Wijesekara et al. [24] recently reported that consumers gave only acceptable scores to the texture and appearance of stirred yogurt with natural phenolics extracted from different plants, while nonfortified yogurt received the highest mean consumer liking score. In another study, Jabuticaba and Jamelão peel powders were added to stirred yogurt, and samples with higher concentrations of powder were less accepted by consumers [34]. Although none of the tannin-added yogurts received higher consumer acceptability than the control yogurt in the present study, it should be considered that the participants were not informed that they were testing a functional yogurt containing tannins with potentially high antioxidant and biological activities. Moreover, knowing that information usually has a positive effect on people’s acceptance and interest in consuming a product.

## 4. Conclusions

The present study investigated the possibility of using wood-extracted tannins in stirred yogurt as a significant source of polyphenols with potentially nutritional activities. To this and the effect of the addition of six food-grade wood tannins on the physicochemical and sensory characteristics of a stirred type of yogurt was evaluated during 21 days of cold storage. The addition of tannins at two concentrations significantly improved the antiradical activity and total phenolic content of the stirred yogurt and increased the syneresis and, to some extent, acidity of the samples with respect to the control. The results revealed that the addition of tannins maintained the viability of the yogurt cultures after 21 days of storage and QS_SOL and FNG/R tannins improved the survival of LAB compared to control. Consumer evaluation showed that yogurt with the lower amount of tannin (0.5%) received higher scores, and, particularly, the yogurts added with 0.5% of FNG/R showed an acceptability similar to control. Based on the obtained results, yogurt enriched with the FNG/R wood-extracted tannin at 0.5% was demonstrated to be effective in terms of producing a novel product with improved functional characteristics according to consumer demands.

## Figures and Tables

**Table 1 foods-12-00405-t001:** Values of pH, acidity (g lactic acid/g yogurt), and syneresis (%) of the control yogurt and yogurt with added tannins during storage and results of variance analysis with Duncan’s test (*p* = 0.05) performed between the tannins and the storage times.

	**Tannin Addition**	**pH**				
		**Day1**	**Day7**	**Day14**	**Day21**	**Significance**
Control	-	4.28 ± 0.11 ^Aa^	4.24 ± 0.04 ^Aab^	4.18 ± 0.03 ^Abc^	4.15 ± 0.01 ^Ac^	**
ARB	0.50%	4.32 ± 0.05 ^Aa^	4.19 ± 0.03 ^ABb^	4.1 ± 0.04 ^ABc^	4.01 ± 0.04 ^Bc^	***
	1%	4.27 ± 0.05 ^Aa^	4.17 ± 0.02 ^ABb^	4.07 ± 0.03 ^Bc^	3.99 ± 0.02 ^Bd^	***
C	0.50%	4.32 ± 0.04 ^Aa^	4.22 ± 0.01 ^ABa^	4.13 ± 0.03 ^ABb^	4.01 ± 0.06 ^Bc^	***
	1%	4.31 ± 0.03 ^Aa^	4.19 ± 0.02 ^ABb^	4.06 ± 0.03 ^Bbc^	3.87 ± 0.07 ^Cc^	***
FNG/R	0.50%	4.28 ± 0.10 ^Aa^	4.24 ± 0.04 ^ABa^	4.14 ± 0.03 ^ABb^	4.09 ± 0.08 ^ABc^	***
	1%	4.26 ± 0.06 ^Aa^	4.21 ± 0.02 ^ABa^	4.11 ± 0.02 ^ABc^	4.03 ± 0.02 ^Bc^	***
NN	0.50%	4.26 ± 0.04 ^Aa^	4.24 ± 0.01 ^Ab^	4.15 ± 0.02 ^ABc^	4.07 ± 0.02 ^ABd^	***
	1%	4.3 ± 0.07 ^Aa^	4.15 ± 0.03 ^Bb^	4.12 ± 0.02 ^ABc^	4.04 ± 0.02 ^Bd^	***
QF	0.50%	4.32 ± 0.04 ^Aa^	4.23 ± 0.02 ^ABb^	4.12 ± 0.02 ^ABc^	4.06 ± 0.02 ^ABd^	***
	1%	4.26 ± 0.03 ^Aa^	4.21 ± 0.01 ^ABb^	4.11 ± 0.02 ^ABc^	4.00 ± 0.05 ^Bd^	***
QS-SOL	0.50%	4.27 ± 0.05 ^Aa^	4.18 ± 0.03 ^ABb^	4.06 ± 0.05 ^Bc^	4.03 ± 0.03 ^Bc^	***
	1%	4.29 ± 0.03 ^Aa^	4.21 ± 0.02 ^ABb^	4.05 ± 0.05 ^Bc^	4.01 ± 0.01 ^Bc^	***
Significance		ns	**	***	***	
	**Tannin Addition**	**Acidity**				
		**Day1**	**Day7**	**Day14**	**Day21**	**Significance**
Control	-	0.61 ± 0.01 ^Id^	0.71 ± 0.05 ^Fc^	0.79 ± 0.03 ^Hb^	0.87 ± 0.02 ^Ga^	***
ARB	0.50%	0.75 ± 0.03 ^EFd^	0.82 ± 0.03 ^Dc^	0.89 ± 0.03 ^DEb^	0.98 ± 0.01 ^EFa^	***
	1%	0.78 ± 0.04 ^BCd^	0.84 ± 0.01 ^Cc^	0.91 ± 0.05 ^Gb^	1.04 ± 0.01 ^Ca^	***
C	0.50%	0.79 ± 0.04 ^Bd^	0.87 ± 0.01 ^Ec^	0.95 ± 0.03 ^Bb^	1.11 ± 0.04 ^Ba^	***
	1%	0.83 ± 0.01 ^Ad^	0.93 ± 0.03 ^Bc^	1.09 ± 0.02 ^Ab^	1.19 ± 0.02 ^Aa^	***
FNG/R	0.50%	0.70 ± 0.01 ^Hd^	0.78 ± 0.03 ^Ac^	0.84 ± 0.01 ^Gb^	0.88 ± 0.01 ^Ga^	***
	1%	0.77 ± 0.01 ^CDEd^	0.82 ± 0.03 ^Dc^	0.86 ± 0.03 ^Fb^	0.89 ± 0.00 ^Ga^	***
NN	0.50%	0.77 ± 0.00 ^CDEd^	0.84 ± 0.01 ^CDc^	0.9 ± 0.03 ^CDEb^	0.97 ± 0.03 ^EFa^	***
	1%	0.77 ± 0.01 ^CDd^	0.85 ± 0.03 ^Cc^	0.94 ± 0.03 ^Bb^	1.03 ± 0.03 ^CDa^	***
QF	0.50%	0.72 ± 0.01 ^Gd^	0.79 ± 0.03 ^Ec^	0.83 ± 0.01 ^Gb^	0.90 ± 0.04 ^Ga^	***
	1%	0.77 ± 0.01 ^CDc^	0.79 ± 0.06 ^Ec^	0.9 ± 0.01 ^CDb^	0.95 ± 0.02 ^Fa^	***
QS-SOL	0.50%	0.73 ± 0.04 ^FGd^	0.81 ± 0.02 ^Ec^	0.88 ± 0.01 ^Eb^	0.97 ± 0.02 ^EFa^	***
	1%	0.76 ± 0.02 ^EDd^	0.84 ± 0.02 ^CDc^	0.91 ± 0.04 ^CDb^	1.01 ± 0.01 ^EDa^	***
Significance		***	***	***	***	
	**Tannin Addition**	**Syneresis**				
		**Day1**	**Day7**	**Day14**	**Day21**	**Significance**
Control	-	20.03 ± 0.76 ^Fa^	19.07 ± 0.13 ^Hb^	17.69 ± 0.05 ^Hc^	16.70 ± 0.14 ^Id^	***
ARB	0.50%	38.72 ± 0.44 ^Ea^	37.30 ± 0.52 ^Ga^	35.91 ± 0.88 ^Fc^	33.42 ± 0.32 ^FGc^	***
	1%	41.76 ± 0.41 ^Ca^	38.96 ± 0.22 ^Fb^	37.49 ± 0.11 ^DEc^	35.63 ± 0.02 ^Cd^	***
C	0.50%	38.93 ± 0.97 ^Ea^	37.53 ± 0.85 ^Gb^	34.94 ± 0.69 ^Gc^	33.1 ± 1.08 ^Gd^	***
	1%	43.43 ± 0.14 ^Ba^	40.95 ± 0.44 ^BCb^	39.83 ± 0.48 ^Bc^	36.77 ± 0.31 ^Bd^	***
FNG/R	0.50%	40.31 ± 0.48 ^Da^	39.07 ± 0.28 ^EFb^	36.6 ± 0.12 ^EFc^	32.94 ± 0.39 ^GHd^	***
	1%	43.25 ± 0.50 ^Ba^	41.51 ± 0.39 ^Bb^	38.85 ± 0.12 ^Cc^	36.05 ± 0.64 ^BCd^	***
NN	0.50%	38.29 ± 0.51 ^Ea^	37.10 ± 0.29 ^Ga^	34.10 ± 0.71 ^Gb^	32.10 ± 0.63 ^Hc^	***
	1%	41.74 ± 0.14 ^Ca^	40.35 ± 0.62 ^CDb^	36.93 ± 0.2 ^Ec^	34.26 ± 0.43 ^EFd^	***
QF	0.50%	40.32 ± 0.35 ^Da^	39.66 ± 0.32 ^DEFb^	36.61 ± 0.06 ^EFc^	34.66 ± 0.17 ^DFd^	***
	1%	43.05 ± 0.25 ^Ba^	40.92 ± 0.11 ^BCb^	37.96 ± 0.71 ^Cdc^	35.68 ± 0.12 ^Cd^	***
QS-SOL	0.50%	41.75 ± 0.08 ^Ca^	39.78 ± 0.58 ^DEb^	37.12 ± 0.73 ^DEc^	35.46 ± 0.16 ^CDd^	***
	1%	44.87 ± 0.06 ^Aa^	42.79 ± 0.311 ^Ab^	41.26 ± 0.04 ^Ac^	38.71 ± 0.01 ^Ad^	***
Significance		***	***	***	***	

The results are represented as means ± standard deviation; *n* = 3. Means followed by different lowercase letters indicate significant difference at *p* < 0.05 among tannins; means followed by different uppercase letters indicate significant difference at *p* < 0.05 between storage times; ** *p* < 0.01, *** *p* < 0.001, ns not significant.

**Table 2 foods-12-00405-t002:** Values of CIELab parameters of yogurts without and with tannins at the beginning and end of storage and results of variance analysis with Duncan’s test (*p* = 0.05) performed between the storage times.

	Tannin Addition	L *			a *			b *		
		Day 1	Day 21	Significance	Day 1	Day 21	Significance	Day 1	Day 21	Significance
Control	-	90.61 ± 0.02 ^a^	90.97 ± 0.05 ^a^	ns	−2.16 ± 0.04 ^a^	−2.09 ± 0.01 ^a^	ns	5.99 ± 0.09 ^a^	6.35 ± 0.03 ^b^	*
ARB	0.50%	77.67 ± 0.16 ^b^	79.17 ± 0.29 ^a^	*	4.32 ± 0.04 ^a^	4.14 ± 0.27 ^a^	ns	14.1 ± 0.05 ^a^	13.95 ± 0.09 ^a^	*
	1%	72.88 ± 0.04 ^b^	74.43 ± 0.37 ^a^	*	5.93 ± 0.01 ^a^	6.04 ± 0.32 ^a^	ns	16.97 ± 0.04 ^a^	17.25 ± 0.09 ^a^	*
C	0.50%	71.35 ± 0.04 ^b^	73.17 ± 0.05 ^a^	*	3.45 ± 0.07 ^a^	3.48 ± 0.02 ^a^	ns	21.36 ± 0.01 ^b^	22.02 ± 0.11 ^a^	*
	1%	67.96 ± 0.08 ^b^	69.36 ± 0.08 ^a^	*	3.59 ± 0.56 ^a^	3.60 ± 0.23 ^a^	ns	22.34 ± 0.12 ^a^	22.65 ± 0.04 ^a^	ns
FNG/R	0.50%	88.95 ± 0.25 ^b^	90.07 ± 0.16 ^a^	*	−0.64 ± 0.10 ^a^	−0.71 ± 0.10 ^a^	ns	9.41 ± 0.39 ^a^	9.27 ± 0.03 ^a^	ns
	1%	84.79 ± 0.10 ^b^	85.94 ± 0.25 ^a^	*	−0.50 ± 0.09 ^a^	−0.48 ± 0.18 ^a^	ns	9.52 ± 0.18 ^a^	9.86 ± 0.24 ^a^	ns
NN	0.50%	78.03 ± 0.02 ^b^	81.12 ± 0.20 ^a^	**	6.04 ± 0.52 ^a^	5.57 ± 0.07 ^a^	ns	16.47 ± 0.53 ^a^	15.84 ± 0.35 ^a^	ns
	1%	74.65 ± 0.70 ^b^	77.78 ± 0.47 ^a^	**	8.18 ± 0.12 ^a^	7.38 ± 0.16 ^a^	ns	18.81 ± 0.11 ^a^	17.89 ± 0.12 ^b^	*
QF	0.50%	80.61 ± 0.09 ^b^	83.16 ± 0.61 ^a^	**	3.96 ± 0.10 ^a^	4.21 ± 0.25 ^a^	ns	10.47 ± 0.33 ^a^	10.29 ± 0.11 ^a^	ns
	1%	75.73 ± 0.12 ^b^	78.57 ± 0.15 ^a^	**	6.01 ± 0.07 ^a^	6.17 ± 0.13 ^a^	ns	11.83 ± 0.02 ^a^	12.46 ± 0.05 ^b^	*
QS-SOL	0.50%	77.56 ± 0.15 ^b^	79.01 ± 0.88 ^a^	*	8.82 ± 0.02 ^a^	8.90 ± 0.02 ^a^	ns	15.65 ± 0.13 ^a^	15.72 ± 0.03 ^a^	ns
	1%	74.21 ± 0.05 ^b^	76.6 ± 0.010 ^a^	*	11.15 ± 0.02 ^a^	10.95 ± 0.02 ^a^	ns	17.85 ± 0.28 ^a^	17.56 ± 0.21 ^a^	ns

The results are represented as means ± standard deviation; *n* = 3. Means followed by different lowercase letters indicate significant difference at *p* < 0.05 between storage times; ns not significant; * *p* < 0.05; ** *p* < 0.01.

**Table 3 foods-12-00405-t003:** Values of TPC (mg GAE/g) and RSA (μM Trolox/g) of the yogurt without and with tannins during cold storage and results of variance analysis with Duncan’s test (*p* = 0.05) performed between the tannins and the storage times.

	**Tannin Addition**	**TPC**				
		**Day1**	**Day7**	**Day14**	**Day21**	**Significance**
Control	-	0.03 ± 0.00 ^Ja^	0.03 ± 0.00 ^Ka^	0.03 ± 0.00 ^Ja^	0.03 ± 0.00 ^Ka^	ns
ARB	0.50%	1.06 ± 0.01 ^Ia^	0.97 ± 0.03 ^Ha^	0.82 ± 0.03 ^Ib^	0.79 ± 0.03 ^Ic^	***
	1%	2.73 ± 0.01 ^Da^	2.54 ± 0.07 ^Db^	1.89 ± 0.03 ^Dc^	1.46 ± 0.12 ^Ed^	***
C	0.50%	1.19 ± 0.03 ^Ha^	1.07 ± 0 ^Jb^	0.78 ± 0.03 ^Ib^	0.49 ± 0.02 ^Gc^	***
	1%	2.24 ± 0.04 ^EFa^	2.01 ± 0.01 ^Eb^	1.41 ± 0.01 ^Fc^	0.97 ± 0.02 ^Fd^	***
FNG/R	0.50%	2.26 ± 0.03 ^Ea^	1.95 ± 0.05 ^Fb^	1.66 ± 0.09 ^Ec^	1.29 ± 0.04 ^Fd^	***
	1%	4.49 ± 0.13 ^Aa^	4.39 ± 0.08 ^Ab^	3.89 ± 0.02 ^Ab^	3.69 ± 0.03 ^ac^	***
NN	0.50%	1.21 ± 0.01 ^Ha^	0.96 ± 0.05 ^Jb^	0.9 ± 0.01 ^Hc^	0.77 ± 0.02 ^Id^	***
	1%	2.17 ± 0.03 ^Fa^	2.06 ± 0.03 ^Eb^	1.84 ± 0.06 ^Dc^	1.6 ± 0.01 ^Dd^	***
QF	0.50%	1.53 ± 0.01 ^Ga^	1.46 ± 0.03 ^Ib^	1.23 ± 0.04 ^Gc^	1.11 ± 0.05 ^Gc^	***
	1%	3.27 ± 0.01 ^Ca^	3.02 ± 0.01 ^Cb^	2.85 ± 0.06 ^Cc^	2.64 ± 0.1 ^Cd^	***
QS-SOL	0.50%	2.22 ± 0.08 ^EFa^	2.09 ± 0.03 ^Gb^	1.92 ± 0.05 ^Dc^	1.6 ± 0.08 ^Bd^	***
	1%	3.98 ± 0.08 ^Ba^	3.79 ± 0.14 ^Db^	3.7 ± 0.04 ^Bc^	3.39 ± 0.04 ^Bc^	***
Significance		***	***	***	***	
	**Tannin Addition**	**RSA**				
		**Day1**	**Day7**	**Day14**	**Day21**	**Significance**
Control	-	0.03 ± 0.00 ^Ia^	0.03 ± 0.00 ^Ka^	0.02 ± 0.00 ^Ga^	0.03 ± 0.00 ^Ga^	ns
ARB	0.50%	5.77 ± 0.44 ^Ga^	4.94 ± 0.58 ^Ib^	3.68 ± 0.07 ^Ec^	3.86 ± 0.23 ^Ec^	***
	1%	9.19 ± 0.13 ^Ca^	9.04 ± 0.25 ^Da^	7.27 ± 0.36 ^Cb^	7.04 ± 0.03 ^Cc^	***
C	0.50%	4.89 ± 0.01 ^Ha^	4.02 ± 0.01 ^Gb^	3.02 ± 0.02 ^Fc^	3.05 ± 0.09 ^Fd^	***
	1%	8.92 ± 0.03 ^CDa^	8.33 ± 0.08 ^Ea^	7.02 ± 0.04 ^Cb^	6.22 ± 0.14 ^Dc^	***
FNG/R	0.50%	8.49 ± 0.04 ^Ea^	7.81 ± 0.13 ^Fb^	7.37 ± 0.04 ^Cc^	6.82 ± 0.11 ^CDd^	***
	1%	17.41 ± 0.38 ^Aa^	15.74 ± 0.14 ^Ab^	14.35 ± 0.06 ^Ac^	12.91 ± 0.51 ^Ad^	***
NN	0.50%	4.84 ± 0.10 ^Ha^	4.32 ± 0.03 ^Gb^	3.18 ± 0.02 ^Fc^	2.95 ± 0.11 ^Fc^	***
	1%	8.67 ± 0.21 ^DEa^	8.19 ± 0.11 ^Eb^	7.26 ± 0.30 ^Cc^	6.29 ± 0.39 ^CDd^	***
QF	0.50%	5.79 ± 0.07 ^Ga^	5.36 ± 0.06 ^Hb^	4.89 ± 0.05 ^Dc^	4.47 ± 0.15 ^Ec^	***
	1%	11.67 ± 0.02 ^Ba^	11.15 ± 0.07 ^Ca^	9.86 ± 0.24 ^Bb^	8.31 ± 0.67 ^Bb^	***
QS-SOL	0.50%	6.81 ± 0.20 ^Ea^	6.55 ± 0.32 ^Ga^	4.98 ± 0.41 ^Db^	4.19 ± 0.94 ^Eb^	***
	1%	11.89 ± 0.02 ^Ba^	11.51 ± 0.04 ^Ba^	9.95 ± 0.21 ^Bb^	8.08 ± 0.73 ^Bc^	***
Significance		***	***	***	***	

Data (mean ± standard deviation; *n* = 3) were expressed as dry weight. Means followed by different lowercase letters indicate significant difference at *p* < 0.05 among tannins; means followed by different uppercase letters indicate significant difference at *p* < 0.05 between storage times; ns not significant; *** *p* < 0.001.

**Table 4 foods-12-00405-t004:** Values (g/100 g) of citric acid, lactic acid, and lactose of the yogurt without and with tannins during cold storage and results of variance analysis with Duncan’s test (*p* = 0.05) performed between the tannins and the storage times.

		**Tannin Addition**	**Storage (Day)**				
			**1**	**7**	**14**	**21**	**Significance**
Citric acid	Control	-	3.13 ± 0.12 ^Ca*^	2.83 ± 0.05 ^ABCb^	2.24 ± 0.01 ^FGc^	2.19 ± 0.01 ^FGc^	***
	ARB	0.5%	3.20 ± 0.04 ^BCa^	2.90 ± 0.12 ^ABCab^	2.88 ± 0.05 ^Cb^	2.83 ± 0.05 ^Cb^	***
		1%	3.24 ± 0.03 ^ABCa^	2.94 ± 0.04 ^ABb^	3.09 ± 0.04 ^Aab^	3.04 ± 0.04 ^Aab^	***
	C	0.5%	3.28 ± 0.03 ^ABCa^	2.86 ± 0.02 ^ABCb^	2.10 ± 0.03 ^Gb^	2.05 ± 0.03 ^Gb^	***
		1%	3.29 ± 0.04 ^ABCa^	2.98 ± 0.03 ^Ab^	2.39 ± 0.09 ^EFb^	2.34 ± 0.09 ^EFb^	***
	FNG/R	0.5%	3.41 ± 0.15 ^Aa^	2.76 ± 0.10 ^ABCDb^	2.28 ± 0.03 ^Fc^	2.23 ± 0.02 ^Fc^	***
		1%	3.35 ± 0.09 ^ABa^	3.01 ± 0.27 ^Ab^	2.70 ± 0.03 ^Dc^	2.65 ± 0.03 ^Dc^	***
	NN	0.5%	3.12 ± 0.11 ^Ca^	2.52 ± 0.26 ^Db^	2.66 ± 0.24 ^Dc^	2.61 ± 0.24 ^Dc^	***
		1%	3.34 ± 0.10 ^ABa^	2.79 ± 0.20 ^ABCb^	2.66 ± 0.05 ^Dc^	2.61 ± 0.05 ^Dc^	***
	QF	0.5%	3.11 ± 0.12 ^ABCa^	2.98 ± 0.09 ^Ab^	2.91 ± 0.03 ^BCb^	2.86 ± 0.03 ^BCb^	***
		1%	3.28 ± 0.22 ^ABCa^	2.98 ± 0.03 ^Ab^	3.05 ± 0.12 ^ABb^	3.00 ± 0.12 ^ABc^	***
	QS-SOL	0.5%	2.94 ± 0.06 ^Da^	2.66 ± 0.12 ^CDb^	2.48 ± 0.02 ^Ec^	2.43 ± 0.02 ^Ec^	***
		1%	3.14 ± 0.15 ^Ca^	2.68 ± 0.09 ^BCDb^	2.78 ± 0.06 ^CDb^	2.73 ± 0.06 ^CDb^	***
Significance			***	***	***	***	
		**Tannin Addition**	**Storage (Day)**				
			**1**	**7**	**14**	**21**	**Significance**
Lactic acid	Control	0	11.82 ± 0.05 ^FGd^	12.82 ± 0.06 ^Cc^	13.84 ± 0.06 ^EFb^	14.25 ± 0.09 ^Ga^	***
	ARB	0.5%	12.29 ± 0.05 ^BCd^	13.78 ± 0.10 ^Bc^	15.23 ± 0.34 ^Bb^	17.08 ± 0.18 ^Ba^	***
		1%	12.40 ± 0.05 ^ABd^	13.65 ± 0.03 ^Bc^	16.14 ± 0.52 ^Ab^	17.17 ± 0.27 ^Ba^	***
	C	0.5%	12.17 ± 0.07 ^Cd^	13.50 ± 0.30 ^Bc^	14.52 ± 0.30 ^CDb^	15.95 ± 0.07 ^CDa^	***
		1%	12.49 ± 0.03 ^Ad^	14.30 ± 0.19 ^Ac^	16.00 ± 0.22 ^Ab^	18.06 ± 0.25 ^Aa^	***
	FNG/R	0.5%	11.67 ± 0.08 ^Gd^	12.89 ± 0.29 ^Cc^	13.91 ± 0.29 ^EFb^	14.92 ± 0.20 ^Fa^	***
		1%	12.02 ± 0.20 ^DEd^	13.06 ± 0.02 ^Cc^	14.08 ± 0.02 ^DEb^	15.18 ± 0.02 ^EFa^	***
	NN	0.5%	12.18 ± 0.16 ^CDd^	12.84 ± 0.14 ^Cc^	13.86 ± 0.14 ^EFb^	15.12 ± 0.28 ^EFa^	***
		1%	12.12 ± 0.10 ^CDd^	12.99 ± 0.20 ^Cc^	14.05 ± 0.14 ^DEb^	15.56 ± 0.19 ^DEa^	***
	QF	0.5%	11.90 ± 0.07 ^EFd^	12.48 ± 0.21 ^Dc^	13.50 ± 0.21 ^Fb^	14.90 ± 0.34 ^Fa^	***
		1%	11.79 ± 0.09 ^FGd^	13.61 ± 0.16 ^Bc^	15.02 ± 0.22 ^BCb^	16.28 ± 0.38 ^Ca^	***
	QS-SOL	0.5%	12.21 ± 0.12 ^BCd^	13.17 ± 0.28 ^Cc^	14.19 ± 0.28 ^DEb^	15.17 ± 0.28 ^EFa^	***
		1%	12.39 ± 0.05 ^ABc^	13.08 ± 0.11 ^Cb^	14.34 ± 0.50 ^DEa^	15.48 ± 0.33 ^Ea^	***
Significance			***	***	***	***	
		**Tannin Addition**	**Storage (Day)**				
			**1**	**7**	**14**	**21**	**Significance**
Lactose	Control	-	37.72 ± 0.99 ^Aa^	32.47 ± 1.22 ^ABb^	26.61 ± 1.03 ^BCc^	23.85 ± 0.41 ^Ad^	***
	ARB	0.5%	37.21 ± 0.64 ^Aa^	33.30 ± 0.89 ^Ab^	26.56 ± 2.20 ^BCc^	21.13 ± 1.06 ^EFd^	***
		1%	37.34 ± 0.49 ^Aa^	30.55 ± 0.54 ^Db^	22.74 ± 1.58 ^Dc^	19.90 ± 0.09 ^Gd^	***
	C	0.5%	34.72 ± 0.82 ^Ca^	28.80 ± 1.00 ^Eb^	25.80 ± 0.99 ^Cc^	20.65 ± 0.64 ^FGd^	***
		1%	35.77 ± 0.95 ^BCa^	27.35 ± 0.45 ^Fb^	22.97 ± 1.67 ^Dc^	18.11 ± 0.49 ^Hd^	***
	FNG/R	0.5%	35.07 ± 0.62 ^Ca^	31.83 ± 0.29 ^BCb^	27.71 ± 0.92 ^ABCc^	23.00 ± 0.41 ^BCd^	***
		1%	36.80 ± 0.88 ^ABa^	31.81 ± 0.18 ^BCb^	27.80 ± 1.06 ^ABCc^	22.51 ± 0.43 ^BCDd^	***
	NN	0.5%	35.89 ± 0.73 ^BCa^	31.96 ± 0.31 ^Bb^	28.65 ± 0.62 ^ABCc^	23.44 ± 0.36 ^Bd^	***
		1%	34.90 ± 0.37 ^Ca^	30.77 ± 0.64 ^CDb^	26.16 ± 2.15 ^BCc^	22.38 ± 0.34 ^CDd^	***
	QF	0.5%	35.07 ± 0.56 ^Ca^	32.16 ± 0.46 ^ABb^	29.16 ± 0.46 ^Ac^	25.81 ± 0.40 ^CDd^	***
		1%	34.71 ± 0.93 ^Ca^	32.23 ± 0.56 ^ABb^	27.85 ± 0.79 ^ABCc^	22.85 ± 0.54 ^CDd^	***
	QS-SOL	0.5%	37.78 ± 0.61 ^Aa^	31.71 ± 0.03 ^BCb^	29.27 ± 0.66 ^Ac^	22.84 ± 0.49 ^BCDd^	***
		1%	36.64 ± 0.50 ^ABa^	32.54 ± 0.55 ^ABb^	27.33 ± 1.52 ^ABCc^	21.97 ± 0.38 ^DEd^	***
Significance			***	***	***	***	

Data (mean ± standard deviation; *n* = 3) were expressed as dry weight. Means followed by different lowercase letters indicate significant difference at *p* < 0.05 among tannins; means followed by different uppercase letters indicate significant difference at *p* < 0.05 between storage times; *** *p* < 0.001.

**Table 5 foods-12-00405-t005:** Value (CFU × 10^6^/g) of lactic acid bacteria (LAB), *Streptococcus thermophilus* (ST), and *Lactobacillus delbrueckii* spp. *bulgaricus* (LD) in yogurt without and with tannins after 21 days of cold storage.

	Tannin Addition	LAB	LD	ST
Control	-	250 ± 1 ^C^	85 ± 6 ^B^	165 ± 7 ^D^
ARB	0.50%	62 ± 5 ^D^	23 ± 3 ^DE^	39 ± 8 ^E^
	1%	86 ± 7 ^DE^	32 ± 2 ^EF^	54 ± 5 ^EF^
C	0.50%	52 ± 8 ^F^	35 ± 6 ^EF^	17 ± 2 ^G^
	1%	14 ± 8 ^G^	6.5 ± 4 ^G^	7.5 ± 5 ^G^
FNG/R	0.50%	285 ± 4 ^B^	115 ± 5 ^A^	170 ± 9 ^CD^
	1%	305 ± 10 ^B^	120 ± 5 ^A^	185 ± 5 ^C^
NN	0.50%	84 ± 8 ^DE^	67 ± 6 ^C^	17 ± 2 ^G^
	1%	71 ± 6 ^EF^	59 ± 3 ^CD^	12 ± 3 ^G^
QF	0.50%	110 ± 11 ^EF^	45 ± 6 ^F^	65 ± 5 ^F^
	1%	86 ± 10 ^DE^	35 ± 5 ^EF^	51 ± 5 ^EF^
QS-SOL	0.50%	310 ± 10 ^B^	105 ± 3 ^A^	205 ± 7 ^B^
	1%	340 ± 17 ^A^	110 ± 7 ^A^	230 ± 10 ^A^
Significance		***	***	***

Means followed by different uppercase letters indicate significant difference at *p* < 0.05 among tannins in columns; *** *p* < 0.001.

**Table 6 foods-12-00405-t006:** Mean values of consumer evaluations of yogurt without and with tannins by the 9-point hedonic scale.

	Tannin Addition	Appearance	Odor	Taste	Flavour	Texture	Overall Liking
Control	-	6.3	5.8	6.3	6.0	6.8	6.0
QF	0.5%	5.0	3.8	2.4	2.5	3.8	2.3
	1%	5.1	4.1	2.3	2.6	3.6	2.3
NN	0.5%	5.8	4.2	2.9	3.3	4.7	3.0
	1%	5.5	5.1	2.6	3.1	3.6	2.8
FNG/R	0.5%	6.7	4.7	4.2	3.8	4.8	3.8
	1%	7.0	3.9	3.7	3.6	4.5	3.5
C	0.5%	3.7	4.5	3.3	3.4	5.5	3.3
	1%	3.3	4.8	2.7	2.9	4.5	2.8
ARB	0.5%	4.9	5.3	4.6	4.3	5.3	4.4
	1%	4.3	3.9	2.3	2.5	4.8	2.6
QS-SOL	0.5%	7.3	5.8	4.1	4.2	5.4	4.1
	1%	7.0	5.9	3.8	4.2	4.9	3.8

## Data Availability

Data is contained within the article.

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
