# Peer review of "Effect of Yogurt Enrichment with Wood Tannin during Shelf Life: Focus on Physicochemical, Microbiological, and Sensory Characteristics"

_foods, 2023, doi:10.3390/foods12020405_

Round 1

Reviewer 1 Report

Author produced the yogurt enriched with various WTEs and then monitor on their physicochemical, microbiological and sensory acceptability during storage for 21 days. Overall, this study is interesting and seem to provide the useful information to milk industry, aimed to increase functional properties of yogurt product. However, some experimental design should be clear and lack of in-depth exploration/discussion in each parameter.

Overall, the article may be acceptable for publication after the authors make some major revisions. Please answers these following questions (the text related with queries are highlighted in yellow):

- Firstly, I suggested that the topic should tell reader that this study monitors the quality changes of yogurt during storage. For example, “Physicochemical, microbiological and sensory characteristics of yogurt enriched with different wood tannins during storage” etc. Please re-think.

ABSTRACT

- Abstract can be improved. The introduction sentence can be more comprehensive (likes is it normal to add fruits to yogurt to improve sensorial quality? // why should polyphenols be added in this product as functional ingredients? Etc.). Moreover, all six sources of tannin should appear. Some empirical results, particularly quantitative data should state to strengthen the abstract.   

- line19-20: All products were accepted by the 19 consumers with an inverse correlation with the percentage of fortification. >>> The data should more specific. What is the kind and level added of them which contained the comparable to original yogurt?  

- I suggested to add other keywords linked to this MS. 3 words seems to be less.

INTRODUCTION

- Please give the ref. of some sentences i.e. line50-51, 52-53, 53-54

- Introduction can improve by these following questions.

(1) Since tannin founds as low amount, some are not edible, particularly tannins from woods. Moreover, it can contribute an astringent or bitter taste to products, therefore, why this study still try to use them by adding in yogurt? Please add the details in the introduction part.

(2) What are six commercial 61 tannin wood extracts (TWE) used in this study. Please provide the details related to them i.e. how different among them? How much tannins contained in them? Etc.

- Moreover, introduction part should end with the objective of the study or give some hypothesis/findings application of the study.

MATERIAL AND METHOD

- Line 72: Do six TWEs used in this study were food grade? Please add this data to ensure their food safety.

-Line79: w/w or w/v or what???

- It seems like author add TWEs after yogurt is fermented (ready to eat)? Why author design like this? Why author did not add them during process?  

- 2.3 is the method for preparation of polyphenol extract, thus it should appear in the beginning of 2.4. Then, for the measurement of DPPH (2.5), author can only mention that extract was prepared as mentioned above.

- 2.6 is the measurement of Organic acids and sugars?

- For 2.7 bacterial count, author should add more details related to sample preparation/dilution. Or add the references of those assay (BAM? USFDA? etc.).

- This study contained sensory evaluation thus the food safety of the experiment products are important. It is certificate/any parameters to ensure the product safety after adding with Twes or during storage for 21 days? Likes pathogen determination, etc.? If had, please add in the MS too.

- Line186-187: Why adding with tannin increased the acidity or decrease pH and how it effect to product? Please add the discussion and compared with some previous studies.

- Line197: Why tannin decrease the syneresis of the product during storage? Please give the scientific discussion related to this result.

- Line206-209: With these discussions, then why author designed to add TWEs when product is already processed? Why not add during processing/before allowing to ferment? This can be suggestion to the further study, please include in any part of this MS.

- Table1,2,3,4,5 please adjust the format for easy reading.

- Table6 why author selected to measure these 3 strains? How it important to yogurt (related with sensorial characteristics or food safety)? And also how it changes after adding with each TWEs. These discussion should add.

-Figure1 is difficult to follow since there are too many samples in this study. Please improved it.

- Important! Now, author did not mention to the various type of tannins used in this experiment (ARB, C, FNG, ……), particularly discussion in details.

CONCLUSION

- It is too general. Please improve it.

- QS_SOL and FNG/R improved LAB and then why? How it improve product quality? and how’s about other parameters after adding with them?

- Novel findings or more specific findings should state. For example, what is the best conditions/suggestion to produce yogurt enriched with tannins? (what things they improved and how the quality changes during storage, compared with traditional product)?

Author Response

We would like to thank the Reviewer for in-depth and improving comments that certainly help our work. A comprehensive list detailing how each point was  addressed and modified according to the suggestions has been reported in the attached file.

Reviewer 2 Report

1. Some of the data should be turned into the images like graphs, which may give a better visual appearance to the content 

2. The fine-tuning of the abstract is very important

3. There is redundancy in the usage of "wood tanning"  and "vegetable tannins". try to use uniform

3. Try to mention the gaps and significance of the study. 

4. Write the previous literature present related to this work like the application of tannin in food

5. Lines 72- 76, mention the scientific name of the plant used for the tannin isolation

6. Use the formula editors for the formula 

7. Mention the references for the methods used in the quality determination of the yogurt

8. Why added the tannins after the yogurt was fermented? why can't at the onset of the yogurt preparation? 

9. Section 2.6 heading should be " Determination of the organic acids and Sugars in yogurt samples"

10. Section 2.8, revise the heading as the " sensory evaluation"

11. The discussion part of the manuscript has to revise a lot

12. In this study there are 3 factors, Tainan type, Tannin concentration, and storage durations. Hence, discussions should be on this basis.

13. The discussion on trend of the results should be discussed effectively. 

14. Table 2, the data of days 1 and 21 only presented? 

15. Why TPC decreased as the storage increased?

16. Mentioned how tannins are added, packed, and stored for the storage studies? 

17. This text "Values (mean ± standard deviation; n = 3)" is presented in the title and footnote of the tables, so, remove it in the heading and mention it in the footnote. 

18. Conclusions should be fine-tuned. 

19. Check for the editorial and typographical issues 

Author Response

We would like to thank the Reviewer for appreciating and greatly improving comments that helped our work. A list explaining in detail how we addressed each comment/suggestion has been prepared for the reviewer

Round 2

Reviewer 1 Report

The authors conducted the required revisions as suggested before.